# Individual Differences in Multisensory Interactions: The Influence of Temporal Phase Coherence and Auditory Salience on Visual Contrast Sensitivity

**Hiu Mei Chow [1,2], Xenia Leviyah [1] and Vivian M. Ciaramitaro [1,*]**

[1] Department of Psychology, Developmental and Brain Sciences, University of Massachusetts Boston, 100 Morrissey Boulevard, Boston, MA 02125, USA; doris1chow@gmail.com (H.M.C.); xleviyah@gmail.com (X.L.)

[2] Department of Ophthalmology and Visual Sciences, University of British Columbia, Vancouver, BC V5Z 3N9, Canada

[*] Correspondence: Vivian.Ciaramitaro@umb.edu; Tel.: +1-617-287-6363

**Abstract:** While previous research has investigated key factors contributing to multisensory integration in isolation, relatively little is known regarding how these factors interact, especially when considering the enhancement of visual contrast sensitivity by a task-irrelevant sound. Here we explored how auditory stimulus properties, namely salience and temporal phase coherence in relation to the visual target, jointly affect the extent to which a sound can enhance visual contrast sensitivity. Visual contrast sensitivity was measured by a psychophysical task, where human adult participants reported the location of a visual Gabor pattern presented at various contrast levels. We expected the most enhanced contrast sensitivity, the lowest contrast threshold, when the visual stimulus was accompanied by a task-irrelevant sound, weak in auditory salience, modulated in-phase with the visual stimulus (strong temporal phase coherence). Our expectations were confirmed, but only if we accounted for individual differences in optimal auditory salience level to induce maximal multisensory enhancement effects. Our findings highlight the importance of interactions between temporal phase coherence and stimulus effectiveness in determining the strength of multisensory enhancement of visual contrast as well as highlighting the importance of accounting for individual differences.

**Keywords:** multisensory perception; visual contrast; phase coherence; temporal coincidence; stimulus effectiveness

---

## 1. Introduction

We live in a multisensory world where we constantly receive information across our different senses. The intricate and tight link of information processing across the senses is demonstrated by a phenomenon called multisensory enhancement, the modulation of information processing of one sense (e.g., vision) by another sense (e.g., audition). For instance, participants perceive the same visual stimulus brighter [1,2], better (i.e., lower detection threshold or higher sensitivity [3–5]) and faster [6], when it is accompanied by a sound versus without a sound, despite the sound bearing no useful task-relevant information. Such effects are important since the enhancement of low-level features (such as brightness or contrast) can potentially enhance the representation of mid-level features, such as object contour and shape completion [7]. An examination of factors contributing to multisensory enhancement provides critical knowledge for understanding basic mechanisms of multisensory processing, as well as its application in areas such as sensory rehabilitation.

Pioneering neurophysiological work on multisensory neurons in the superior colliculus (e.g., [8]) as well as functional neuroimaging work on multisensory cortical areas, such as the superior

temporal sulcus [5,9,10], have identified important stimulus-related factors contributing to multisensory integration. What is collectively agreed upon is that stronger neural responses are often reported when the input from individual senses are spatially aligned (spatial coincidence [11]), temporally synchronous (temporal coincidence [12]), or weak (inverse effectiveness principle [13]). These factors also enhance behavioral measures of sensory processing, although they may not be mandatory for enhancement, as shown for spatial coincidence by Spence [14], and for inverse effectiveness by Holmes [15]. For example, luminance detection is enhanced when audiovisual stimuli are aligned spatially and temporally [16], multisensory facilitation in response time is the greatest behaviorally and neuronally with weak stimulus effectiveness [17] (but see [18] which reported increased facilitation with increased stimulus intensity), and multisensory speech perception shows the greatest enhancement with unisensory stimuli of medium effectiveness (e.g., [19]).

Traditionally, the three predominant factors influencing multisensory interactions have been studied independently. For example, changing stimulus-onset asynchrony while maintaining spatial relations across different sensory stimuli or changing spatial discrepancy while maintaining temporal relation. More recently, studies have examined how these factors interact. For example, interactions between temporal and spatial factors have been examined [9], as well as interactions between temporal and/or spatial factors and stimulus effectiveness [20,21]. For the latter, Fister and colleagues [20] reported that observers asked to judge if audio-visual stimuli were synchronous or not were more tolerant of temporal offsets and showed the strongest multisensory facilitation in response time when the paired stimuli were less effective, suggesting an interaction between temporal coincidence and effectiveness. Similarly, in a localization task, the largest multisensory gain in behavioral performance (accuracy and response time) was observed when weak audiovisual stimuli were presented at peripheral locations [21], suggesting an interaction between spatial coincidence and stimulus effectiveness. What remains to be addressed is whether these interactions between stimulus-related factors are limited to benefits in response time, or also allow for other types of multisensory enhancement, for example, improved visual contrast sensitivity as found in previous studies examining a single factor in isolation [3–5,16].

Here we examined how temporal coincidence and stimulus effectiveness of a task-irrelevant sound influence visual contrast sensitivity in a two-alternative forced-choice psychophysical paradigm. We altered temporal coincidence by manipulating temporal phase coherence between visual and auditory stimuli (i.e., the sound is modulated in-phase vs. out-of-phase with the visual target). We altered stimulus effectiveness, salience, by presenting the sound at five different intensities, unlike most studies including only two intensities, high and low (e.g., [20,21]). We expected the strongest enhancement of visual contrast sensitivity for a visual target changing in phase with a task-irrelevant sound that is only mildly effective.

## 2. Materials and Methods

### 2.1. Participants

Participants were undergraduate or graduate students recruited from the University of Massachusetts Boston community. Our experiment protocol, study number 2013147, was approved by the UMB Institutional Review Board and complied with the Declaration of Helsinki. Written informed consent was obtained from all participants. All participants had no known psychiatric or neurological history, had normal or corrected-to-normal vision, and did not report any hearing problems. Participants were paid $10 per hour or received extra-credit for an approved psychology course.

A power analysis performed in G*Power 3.1.9.2 [22] determined that a sample size of 17 participants was required to yield a power of 0.85 to detect a main effect in a repeated measures ANOVA (effect size $f = 0.3$) with five measurements (five levels of auditory salience). Twenty adults (18–27 years of age, 20 female) participated in the study. Data from two participants were discarded due to outlying performance yielding a total of 18 participants in the current sample.

*2.2. Apparatus and Stimuli*

　　Visual and auditory stimuli were generated and presented by MATLAB R2014b (The MathWorks, Inc., Natick, MA, USA) and Psychtoolbox [23–25] (version: 3.0.12 beta). Visual stimuli were displayed against a uniform gray background (60 cd/m$^2$) on an LCD monitor (Tobii TX300) positioned 60 cm from the participants. Auditory stimuli were presented to the right and left of monitor center (55 cm behind the monitor and 35 cm to each side of the monitor center) via speakers (JA Audio B3-HTPACK) (Figure 1A). Eye position of both eyes was recorded at a sampling rate of 300 Hz via a Tobii eye tracker (TX300).

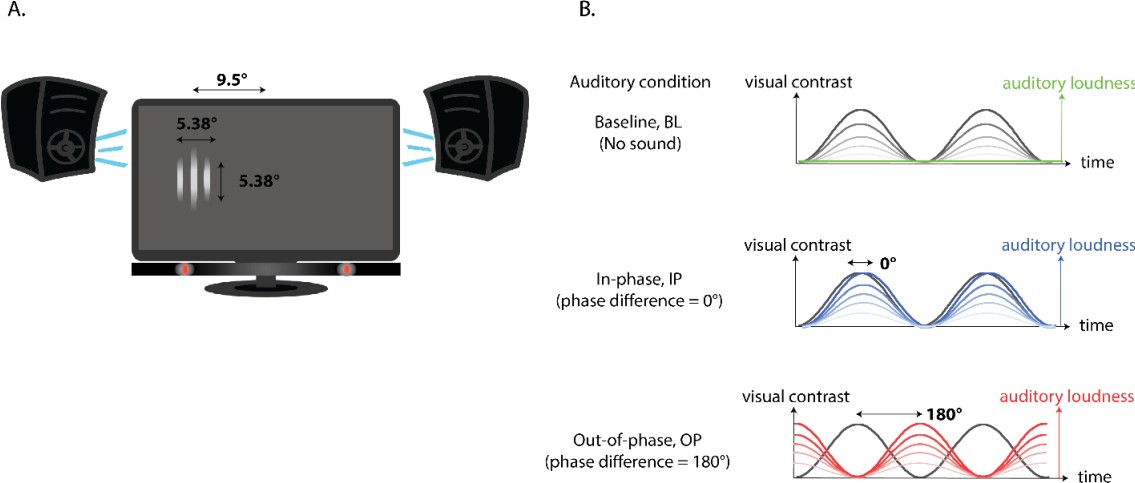

**Figure 1.** (**A**) Experimental set-up, including an illustration of a sample visual target displayed to the left of screen center. (**B**) An illustration of the modulation in visual contrast for a single visual stimulus (black/grey line) and the range in modulation of auditory loudness (baseline condition: green; in-phase condition: blue; out-of-phase condition: red) across time.

　　The visual target was a Gabor patch (diameter = 5.38°, spatial frequency = 0.5 cycles/°; number of cycles = 2.69), presented to the left or right of monitor center at an eccentricity of 9.5°, with visual target location chosen pseudo-randomly from trial to trial. The contrast of the visual target modulated sinusoidally at 1Hz from 0% (the same luminance as monitor background, i.e., invisible) to one of five maximum contrast levels, as indicated by the different shades of grey in Figure 1. The maximum visual contrast was chosen pseudo-randomly from trial to trial. The five contrast levels were selected from a list of seven possible values (0.08%, 0.16%, 0.31%, 0.63%, 1.25%, 2.5%, or 5%) to account for individual differences based on participants' performance during the practice block. Visual contrast difficulty level was set by default to the medium range (0.16–2.5%), and made harder or easier after a practice block of 30 trials (10 trials for each contrast: lowest, middle and highest) to ensure performance was close to 50% for the lowest contrast level and close to 90–100% for the highest contrast level. Three participants completed the experimental blocks at the hardest contrast range (0.08–1.25%), 15 at the medium range (0.16–2.5%), and 2 at the easiest range (0.31–5%).

　　The auditory stimulus was the same amplitude-modulated white noise sound presented via both speakers against a constant background white noise of 34 dB. The auditory stimulus was produced by multiplying a modulating wave (modulating frequency = 1 Hz) and a constant white noise sound. The amplitude of the modulating wave determined the loudness, or the auditory salience, of the white noise over time, which varied from 0 dB (no sound against the background noise) to one of five maximum sound levels (31.5, 35.5, 39.5, 47.5, or 55.0 dB), as indicated by the different shades of blue or red in Figure 1. The maximum sound level was chosen pseudo-randomly from trial to trial. In addition to auditory salience, we also manipulated the phase coherence/phase angle difference between the contrast modulation of the visual target and the loudness (amplitude) modulation, of

the auditory stimulus, such that modulation of the auditory stimulus was in-phase (phase angle difference = 0°, starting from minimum loudness, blue lines in Figure 1B), or out-of-phase (phase angle difference = 180°, starting from maximum loudness, red lines in Figure 1B), with contrast modulation of the visual target.

## 2.3. Procedure

This experiment adopted a two-alternative forced-choice paradigm to probe visual contrast threshold based on participants' eye movement response (left or right). This experiment was conducted in a dark, sound-proof, chamber to minimize ambient noise from the environment. A five-point calibration was used to calibrate the eye tracker.

In a given trial, participants first saw a fixation point at monitor center. To capture participants' attention, the fixation point changed either in shape, color, size, or angle every 150 msec. For example, when the fixation point changed in size, the width and height of the fixation point could vary between 0.913° and 2.931°; or when the fixation point changed in shape, the fixation point could switch between five types of geometric shapes: cross, circle, triangle, star, and heart-shape. Participants triggered the stimulus display by directing their gaze to screen center (fixation window: 2.536° × 2.536°, minimum fixation period: 1 sample = 3.33 msec). Then, the fixation point disappeared and the visual target appeared to the left or right of monitor center in one of the three auditory conditions: in-phase (IP), out-of-phase (OP), or without sound (baseline, or BL). Participants were asked to look towards the visual target when they detected it and had to respond within 20 s or the trial would be discarded from analysis and repeated. The first eye position sample where the participants' horizontal eye position went beyond the 8.063 deg from the center while their vertical eye position maintained within +/− 4.607 deg from the horizontal midline was considered a response. Following the eye movement response, participants received feedback. For both correct and incorrect responses, a black box outline (5.38° × 5.38°) appeared for 1 sec where the visual stimulus had been presented. To promote orienting towards the visual target, a short cartoon video (5.38° × 5.38°) played for 1 sec inside the black box for correct responses.

In total, there were 11 auditory conditions (2 phase coherences [IP or OP] × 5 auditory salience levels + no sound control [BL]). Participants completed 100 trials for each auditory condition (20 trials per contrast × 5 visual contrasts) yielding a total of 1100 trials. Two repeats of each visual and auditory condition were selected at random for each block of 110 trials, for a total of 10 blocks, completed in two sessions of 1 hour each, one session per day.

Prior to starting the experiment, participants completed a practice block of 30 BL trials, which included a visual target at low, medium, and high contrast, but no auditory stimulus. This was to familiarize participants with the stimuli and procedures and to assess if the difficulty of the visual contrast range was appropriate (see Apparatus and Stimuli).

## 2.4. Data Analysis

**Psychometric function fitting:** Percent correct for each visual contrast and auditory condition was computed for each observer. Data were fit with a Weibull function using MATLAB R2014b (The MathWorks, Inc., Natick, MA, USA) and Psignifit 4.0 [26] to derive the 75% threshold. Figure 2 plots the psychometric functions across conditions from a representative participant.

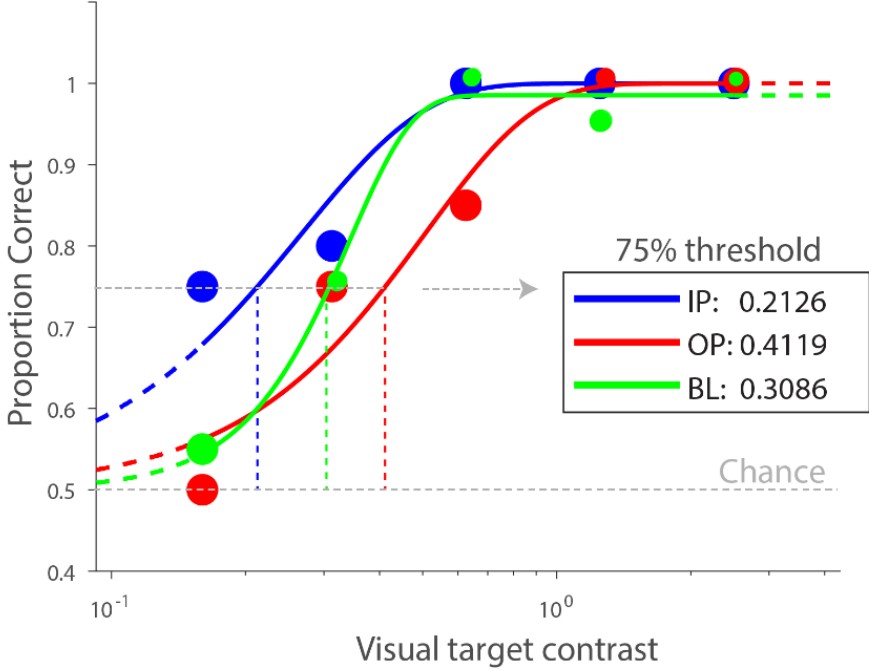

**Figure 2.** Psychometric functions of baseline (BL), in-phase (IP) (mid-salience), and out-of-phase (OP) (mid-salience) conditions from a representative participant showing the expected effect: estimated visual contrast threshold is the lowest for IP, then BL, then OP condition. The size of individual dots is adjusted so that overlapping data points are more visible.

**Threshold difference measures:** For each comparison of interest, we computed a difference measure between threshold values. To quantify the expected facilitative effect on visual contrast performance of the IP relative to the no sound condition, we computed the difference between the log of IP threshold and the log of BL threshold (BL-IP), for each auditory salience level. A positive threshold difference would indicate that the IP sound reduced threshold relative to baseline, an IP benefit, while a negative threshold difference would indicate an IP detriment. One sample *t*-tests corrected for multiple comparisons by False Discovery Rate (FDR) were used to evaluate if the difference measures were significantly different from zero using R Studio [27] (version 1.1.423) and *t.test* and *p.adjust* functions. Similarly, we quantified the *expected suppressive effect* of the OP relative to no sound condition (OP-BL), as well as the *expected facilitative effect* of IP relative to OP condition (OP-IP), for each auditory salience level. In addition to statistical test results, we also reported the upper and lower limit of the 95% confidence interval of the threshold difference, and Hedges' *g*, an effect size measure calculated and interpreted similarly to Cohen's *d* but corrected for small sample sizes.

Based on the known multisensory effects of temporal coincidence and inverse effectiveness, we expected the strongest facilitative effects for the IP relative to the BL and OP conditions, at a mid-auditory-salience (Figure 3A), with diminishing, or even reversing effects, at an extreme auditory salience (too soft or too loud). This was evaluated using a one-way repeated measures Analysis of Variance (ANOVA), performed by R Studio and *ez* package [28], to examine the effect of auditory salience on the magnitude of difference measures for each of our 3 comparisons.

Similar steps were followed to calculate slope differences, without taking the log of slope estimates. For reaction time differences, we excluded trials for which the reaction time was 2 *SD* slower than the participant's average reaction time across trials of the same condition and auditory salience. We then computed a mean reaction time for each condition and auditory salience, for each participant. Then we computed reaction time differences based on the mean reaction time for each condition-of-interest, for each participant. Given that we did not design our paradigm to optimize the speed of response,

i.e., this is not a speeded reaction time task, we provide these results in Supplementary Materials for interested readers (Figures S1 and S2; Tables S1 and S2).

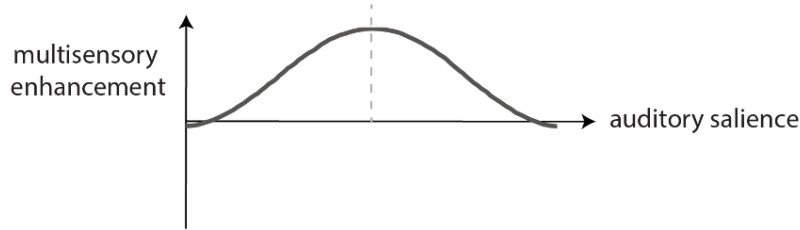

A. Expected results: Maximum multisensory enhancement at mid-salience

B. Possible sources of individual differences

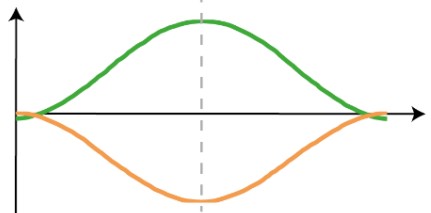

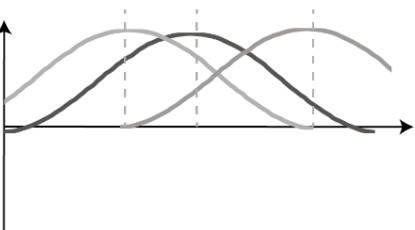

**Figure 3.** (**A**) Illustration of hypothetical data showing the expected results where multisensory enhancement peaks at mid-auditory salience, reflecting the interaction between temporal phase coherence and auditory salience. (**B**) Illustration of possible sources of individual differences which could arise from the type of multisensory effects, enhancement or suppression, at mid-auditory salience (left), or the optimal auditory salience required for multisensory enhancement (right).

**Exploratory analyses to account for individual differences:** We hypothesized that there could be two possible sources of individual differences (Figure 3B). First, while most previous literature suggests that multisensory integration leads to benefits in performance, it has also been reported to impair performance (e.g., crossmodal distraction [29]). A preliminary analysis of the threshold difference between IP and BL at mid-level auditory salience revealed a clear bimodal distribution of our data into two groups: in one group the IP sound improved contrast threshold relative to no sound (as expected) while the opposite was true for the other group. This observation motivated an exploratory analysis. Participants were grouped into either (1) the enhancement group (positive BL-IP threshold difference) or (2) the suppression group (negative threshold BL-IP difference), based on the sign of difference measures at mid-level auditory salience (39.5 dB). One sample *t*-tests, adjusted for multiple comparisons with false-discovery rate method, were performed to see if each difference measure was significantly different from zero for each group. We expected that, despite the group differences in the direction of multisensory effects (enhancement versus suppression), both groups would show the maximal multisensory effect at mid-level salience (Figure 3B).

Second, previous literature has suggested that the optimal stimulus salience level for multisensory integration could vary across individuals [30]. It is possible that different observers would show maximal multisensory enhancement at different auditory salience levels. To account for such individual differences, we computed the mean threshold difference and performed statistical analysis based on data aligned to each observer's optimal salience level divided into three parts: optimal salience (where the maximum BL-IP threshold difference is observed), optimal salience +/− 1 bin (the salience bin(s) adjacent to the optimal salience level), and optimal salience +/− 2 bins (the salience bin(s) 2 steps

away from the optimal salience level). Data were averaged if two bins were involved. For example, if salience level 3 was optimal, we computed threshold differences for optimal salience +/− 1 bin by averaging salience levels 2 and 4, and threshold differences for optimal salience +/− 2 bins by averaging salience levels 1 and 5. We expected the strongest multisensory effects (i.e., IP benefit relative to BL, OP suppression relative to BL, IP benefit relative to OP) at the optimal salience level with reduced multisensory effects at neighboring saliences, as revealed by a significant main effect of auditory salience in a repeated-measures ANOVA.

## 3. Results

### 3.1. Data Exclusion

Prior to statistical testing, we determined exclusion criteria to account for outlying performance, informed by the graphical distribution of the visual contrast thresholds and threshold differences between conditions. To account for individual differences in perceptual sensitivity to visual stimuli, we excluded participants with poor performance in the visual-only baseline condition, defined as having a BL threshold 2 *SD* from the group mean (mean log (baseline threshold) = −0.352, *SD* = 0.188). Based on this criterion, one participant was excluded. To account for individual variability in multisensory effects, we excluded participants with threshold differences (BL-IP) at the mid-level auditory salience (39.5dB) 2 *SD* from the group mean (mean threshold difference = −0.024, *SD* = 0.163). Based on this criterion, one participant was excluded. These exclusion criteria were not determined prior to data collection as we did not expect to see huge outliers in healthy young participants.

### 3.2. Multisensory Interactions without Accounting for Individual Differences

The average threshold difference for each auditory salience level and each comparison of interest (left: the facilitative effect of IP sound relative to BL; middle: the suppressive effect of OP sound relative to BL; right: the facilitative effect of IP relative to OP sound) is plotted across participants in Figure S1. We did not find any consistent effects of sound on any performance measure (Table S1).

### 3.3. Multisensory Interactions in Individuals Showing Facilitated Visual Processing by IP Sound

We plotted the average threshold difference across multisensory comparisons for participants who showed an enhanced visual contrast threshold for mid-salience IP sounds relative to BL in Figure 4 (top panel). We observed a significant facilitative effect of IP sound relative to BL in four of the five auditory salience levels (except the softest level), FDR-adjusted *p*s < 0.0124, despite the grouping being based on only the mid-level salience (39.5 dB). This suggests, for this group of participants, IP sounds enhanced performance regardless of auditory salience levels. This group of participants also showed a significant benefit of OP sound relative to BL at an adjacent auditory salience level (47.5 dB) ($t(8) = 4.838$, FDR-adjusted $p = 0.006$, Hedges' $g = 0.569$, 95% CI (−0.450, 1.589)). We did not find any effect of OP vs. IP after adjusting for multiple comparisons. Individual trend lines of threshold differences as a function of auditory salience (Figure 4, bottom panel) showed that the facilitative effect of IP sound relative BL is relatively consistent across salience levels and individuals, whereas the other comparisons are variable across individuals.

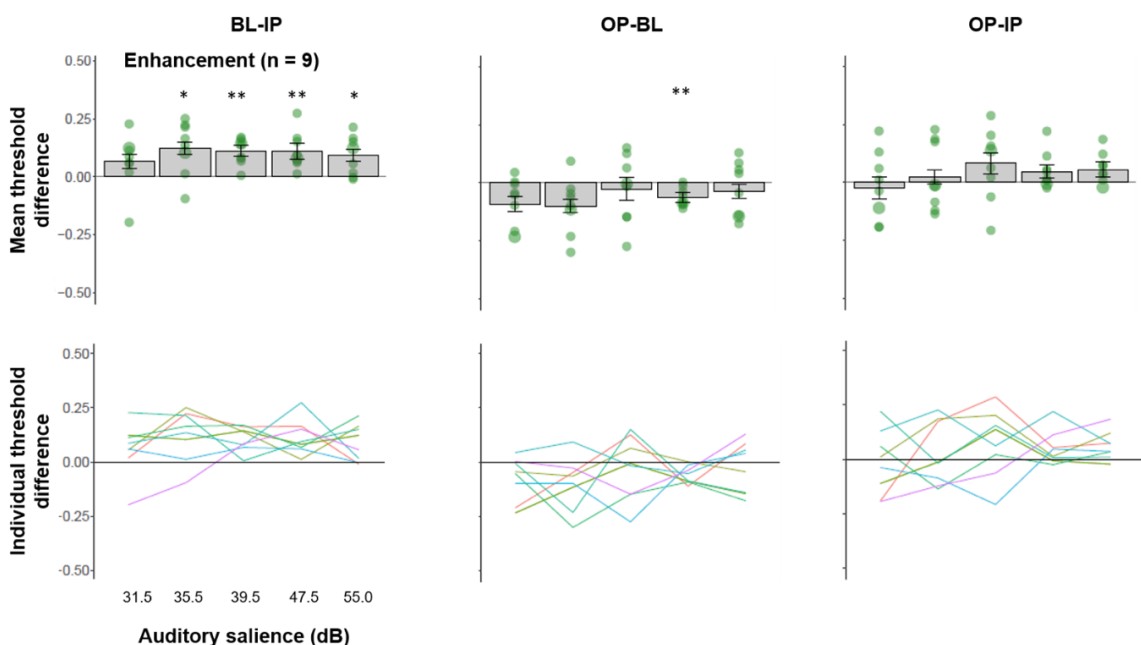

**Figure 4.** (**Top panel**) Mean threshold difference (+/− SE) and (**bottom panel**) trends line for individual participants across three comparisons from the facilitation group. **\*\***: *p* < 0.01, **\***: *p* < 0.05.

### 3.4. Multisensory Interactions in Individuals Showing Suppressed Visual Processing by IP Sound

We plotted the average threshold difference across multisensory comparisons for participants who showed a suppressed visual contrast threshold for mid-salience IP sounds relative to BL in Figure 5 (top panel). We did not find significant threshold differences for any comparisons for the other salience levels (FDR-adjusted *p*s > 0.106) other than the grouping salience level (*t*(8) = 5.617, FDR-adjusted *p* = 0.003, Hedges' *g* = −1.783, 95% CI (−2.965, −0.602)). The significant IP-BL threshold difference at mid-salience level is presumably due to the grouping. Furthermore, at the same salience level, we did not observe an effect of OP sound relative to baseline (*t*(8) = 1.333, FDR-adjusted *p* = 0.366, Hedges' *g* = −0.423, 95% CI (−1.434, 0.587)), or an effect of IP sound relative to OP sound (*t*(8) = 1.166, FDR-adjusted *p* = 0.347, Hedges' *g* = 0.370, 95% CI (−0.638, 1.378)). Individual trend lines of threshold differences as a function of auditory salience (Figure 5, bottom panel) showed that the suppressive effect of IP sound relative BL is relatively consistent across salience levels (especially 35.5, 39.5, 47.5 dB) and individuals, whereas the other comparisons are variable across individuals.

### 3.5. Multisensory Interactions Accounting for Individual Differences in Optimal Auditory Salience

We noticed large individual differences for which auditory salience level allowed the maximal IP relative to BL benefit. While we hypothesized that the IP benefit would peak for auditory stimuli at mid-salience levels (39.5 dB in this study), it is possible that individual participants might have slightly better or worse hearing sensitivity which could shift the peak to a lower or higher salience level. To account for this, we defined, for each observer, the optimal salience level where the maximum BL-IP threshold difference was observed across the five salience levels measured (Table 1). Results for slope and RT differences were plotted and summarized in Figure S2 and Table S2 for reference.

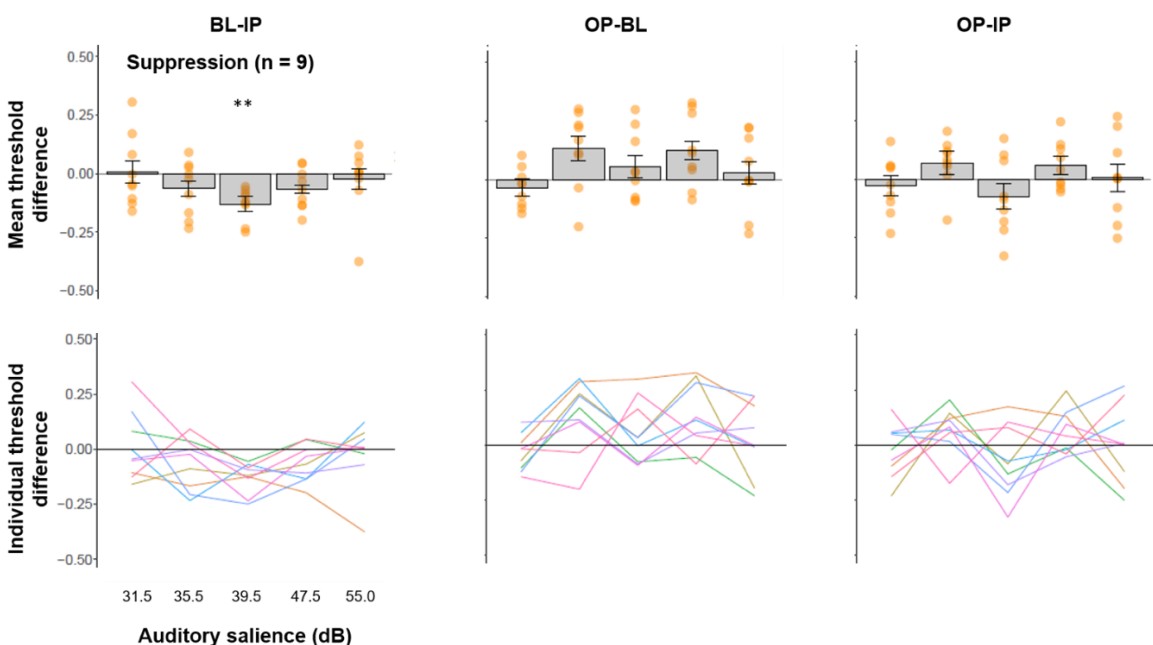

**Figure 5.** (**Top panel**) Mean threshold difference (+/− SE) and (**bottom panel**) trends line for individual participants across three comparisons from the suppression group. **: $p < 0.01$.

**Table 1.** The number of participants showing peak IP benefit relative to BL at each auditory salience level.

| Optimal Salience (Salience with Greatest BL-IP Threshold Difference) | 31.5 dB | 35.5 dB | 39.5 dB | 47.5 dB | 55.0 dB |
|---|---|---|---|---|---|
| Number of participants | 5 | 4 | 3 | 2 | 4 |

The mean threshold difference across participants is plotted in Figure 6 (top panel). We found a significant benefit of IP relative to BL at the optimal salience level (mean threshold difference = 0.135, 95% CI (0.086, 0.184), $t(17) = 5.3921$, FDR-adjusted $p = 0.002$, Hedges' $g = 1.243$). This IP benefit relative to BL is not present at other salience bins (FDR-adjusted $ps \geq 0.555$). This is further confirmed by a significant main effect of salience on the magnitude of BL-IP threshold differences, ($F(2, 34) = 22.157$, $p < 0.001$, $\eta^2_G = 0.281$). Post-hoc analysis revealed that the BL-IP threshold difference at the optimal salience is significantly greater than that of optimal salience +/− 1 bin, (FDR-adjusted $p < 0.001$), and that of optimal salience +/−2 bins (FDR-adjusted $p < 0.001$). The magnitude difference between optimal salience +/− 1 bin and +/− 2 bins is not significant (FDR-adjusted $p = 0.18$).

We also computed OP-BL and OP-IP threshold differences across participants aligned to the optimal salience, as determined by the peak IP benefit relative to BL. We found a significant OP benefit relative to BL at the optimal salience level (mean threshold difference = −0.06, 95% CI (−0.107, −0.022), $t(17) = 3.002$, FDR-adjusted $p = 0.024$, Hedges' $g = -0.692$), but not at other bins, FDR-adjusted $ps \geq 0.678$. However, the main effect of salience on OP-BL threshold difference is not significant, $F(2,34) = 2.777$, $p = 0.076$, $\eta^2_G = 0.091$. We also found a significant IP benefit relative to OP at the optimal salience level (mean threshold difference = 0.083, 95% CI (0.030, 0.135), $t(17) = 3.077$, FDR-adjusted $p = 0.021$, Hedges' $g = 0.709$). This effect is not present at the other salience bins (FDR-adjusted $ps \geq 0.262$). This is further supported by a significant main effect of salience on OP-IP threshold difference ($F(2,34) = 3.774$, $p = 0.033$, $\eta^2_G = 0.127$). Post-hoc analysis revealed that the IP benefit relative to OP is significantly larger than that at the optimal salience +/− 2 bins (FDR-adjusted $p = 0.009$), but not with the optimal salience +/− 1 bin (FDR-adjusted $p = 0.239$).

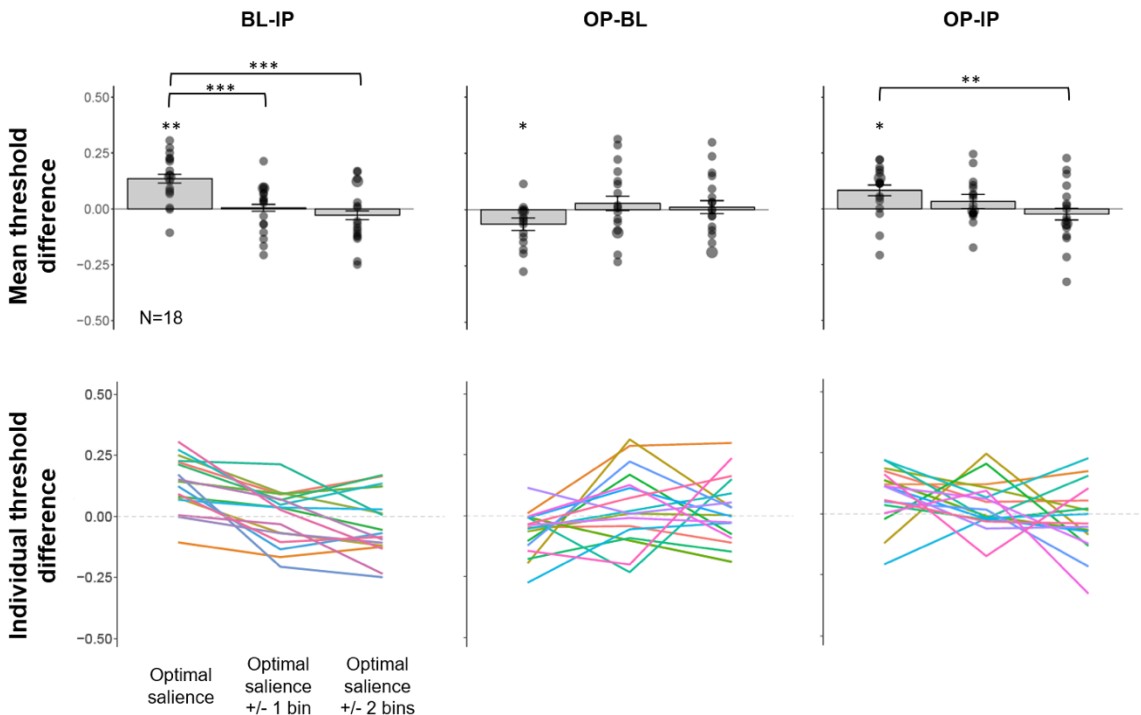

**Figure 6.** (**Top panel**) Mean threshold difference (+/− SE) and (**bottom panel**) trends line for individual participants across three comparisons aligned to the individual observers' optimal salience bins. ***: $p < 0.001$, **: $p < 0.01$, *: $p < 0.05$.

### 3.6. The Relationship between Unisensory Visual Performance and Multisensory Effects

Given the great variability across individuals in our task, we speculated that visual contrast sensitivity at baseline might play a role in determining the strength and direction of the multisensory effects we observed. This was motivated by previous studies showing multisensory benefits are more robust in individuals with visual impairments [31]. If so, we expected to see stronger multisensory effects in individuals with poorer visual performance, larger visual contrast thresholds, in the BL condition. However, we did not find any significant relationship between a participant's unisensory visual performance (log BL threshold) and their multisensory effects (threshold difference estimates) in any comparison, whether effects were estimated at the mid-salience level ($ps > 0.446$, Figure 7 top panel), or based on the individually-determined optimal salience level ($ps > 0.375$, Figure 7 bottom panel).

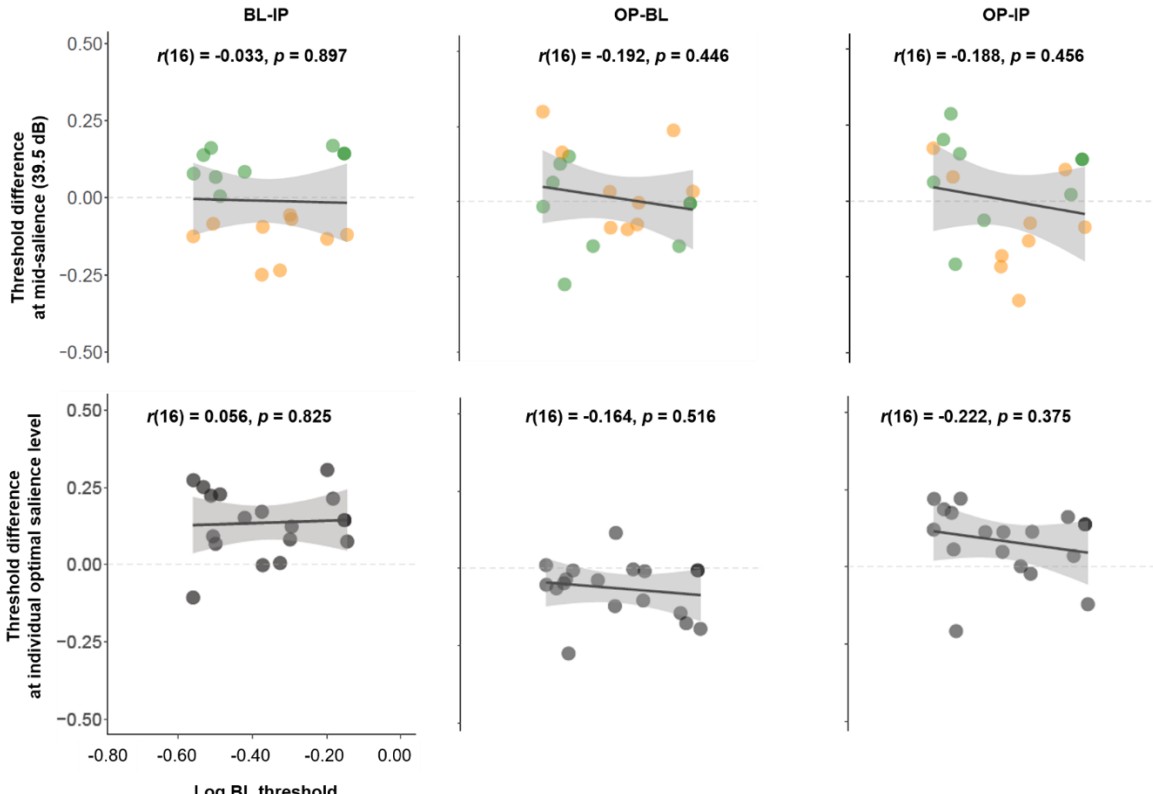

**Figure 7.** Correlation of log BL threshold against threshold differences (BL-IP, OP-BL, and OP-IP) at mid-salience levels (39.5 dB, **top panel**) and at optimal salience levels (**bottom panel**).

## 4. Discussion

This study aimed to explore the interaction between temporal coincidence and stimulus effectiveness, or salience, on multisensory enhancement of the visual contrast of a Gabor pattern by a task-irrelevant sound. We found huge individual differences in the direction of multisensory effects (enhancement vs. suppression) and the optimal auditory salience supporting multisensory enhancement. After aligning the data based on individual observers' optimal auditory salience, we found the expected multisensory enhancement of visual contrast (reduced visual contrast threshold) by a white-noise sound modulated in-phase with the modulation of the visual target, relative to no sound or a sound modulated out-of-phase. More importantly, the magnitude of this multisensory enhancement decreased as auditory stimulus salience moved further away from the optimal salience level. Our results provide direct evidence supporting the interaction of temporal coincidence and stimulus effectiveness in determining the strength of multisensory enhancement of visual contrast and highlight the importance of individual differences.

Our finding adds to the recent literature highlighting interactions between two or more stimulus-related factors in multisensory integration in human behavioral studies (e.g., [9,20,32]) and non-human animal physiological studies [33,34]). A particularly relevant study, by Fister and colleagues [20], showed interactions between the same two factors (temporal coincidence and stimulus effectiveness) examined in our study. In their study, they found the temporal binding window for audio-visual synchrony judgments widened, i.e., a larger stimulus-onset asynchrony was tolerated, in a task where participants attended to both visual and auditory stimuli and reported their temporal relationship. Our results suggest that interactions between temporal coincidence and stimulus effectiveness can also manifest in a different type of multisensory phenomenon (enhanced visual contrast by a sound), a different task (visual target localization), and without explicit top-down attention to auditory information as our sound was task-irrelevant and non-informative. This supports the

notion that interactions between stimulus-related factors are general, and not constrained to specific exemplars of multisensory phenomenon.

In addition, our findings highlight important aspects related to stimulus effectiveness. We did not find evidence for multisensory enhancement when averaging data across participants without accounting for individual differences in the auditory salience optimal for a given observer. In fact, using a pre-set auditory salience, not one unique for each participant, while half of observers showed the expected enhancement of visual contrast by IP sounds, the other half showed impaired visual contrast perception by IP sounds. Most previous studies studying the principle of inverse effectiveness have incorporated only two levels of stimulus effectiveness (high vs. low) (e.g., [35]). Employing a limited sample of stimulus salience levels may be insufficient to fully unravel multisensory processing for two reasons. First, the use of a single stimulus setting across participants does not account for individual differences in the optimal stimulus, potentially washing out multisensory effects when averaging across participants. Second, multisensory effects might not change linearly with increasing stimulus effectiveness. On the contrary, the magnitude of multisensory effects has often been shown to follow an inverted-U shape, with maximal effects observed at medium levels of stimulus effectiveness (e.g., [19]). Such an inverted-U shape function may be lost by having only two levels of stimulus effectiveness, not allowing a meaningful prediction for how stimulus effectiveness alters the magnitude of multisensory processing.

### 4.1. Individual Differences in Multisensory Enhancement

Our results showed individual differences in the magnitude (after aligning the data by the optimal auditory salience levels) and in the direction (at the mid-salience level without alignment) of the multisensory effect. It is important to note that these results are based on exploratory analysis, to be confirmed by future studies where these classification methods of participants and/or stimulus intensity based on performance are pre-determined. Nonetheless, our results are in line with previous studies in multisensory research regarding the effect of individual differences. Below we discuss possible sources of individual differences.

First, the strength of multisensory effects might be determined by unisensory sensitivity. Previous research has reported multisensory enhancement of visual perception by sounds only in low-vision individuals [31,36], or when the visual stimulus occurred in the affected visual field of low-vision individuals [36]. We explored this possibility by examining the correlation between the magnitude of multisensory effect and the visual contrast threshold at baseline (visual only) condition and found no relationship. This rules out individual differences in unisensory visual sensitivity as a contributing factor to individual differences in multisensory effects in healthy sighted individuals. Alternatively, it is possible that individual differences in auditory sensitivity might have contributed to multisensory effects. For instance, individuals with poorer auditory sensitivity might show stronger multisensory effect. This hypothesis cannot be ruled out in our study as we did not measure unisensory auditory performance in our participants.

Additionally, other research has suggested multisensory processing could be influenced by sensory dominance. For example, Giard and Perronnet [37] reported that multisensory presentation of stimuli leads to increased neural activity in the cortex of the weaker unisensory modality. In other words, neural activity in auditory cortex is enhanced by multisensory stimulation in visually dominant participants, and that of visual cortex is enhanced in auditory dominant participants. This is further supported by Romei and colleagues [38] who showed that the multisensory effect was more dependent on the preferred sensory modality than on unisensory sensitivity, as described above. To contextualize these findings in our study, it is possible that participants who showed stronger multisensory effects were auditory dominant whereas participants who showed little or no multisensory effects were visual dominant (in these participants, the reduced multisensory effect might be because they effectively ignored the auditory stimulus). Future research is required to further our understanding of the role of sensory dominance in determining the direction and magnitude of multisensory effects.

Last but not least, individual differences in multisensory effects might be contributed by individual differences in structural and functional differences in the multisensory neural network. This is supported by previous research showing greater multisensory reaction time facilitation in individuals with (1) increased white-matter connectivity between parietal regions and early sensory areas [39] and (2) stronger correlation between neural activity across clusters of the cortical network spanning occipital, parietal and frontal areas [40]. Furthermore, individual differences in the peak and power of alpha frequency neural oscillations have been shown to be related to inter-individual as well as intra-individual trial-by-trial experience in the flash-beep illusion, a multisensory experience where an illusory visual flash is induced by a sound [41]. We encourage future work to examine factors contributing towards individual differences of multisensory enhancement effects, including establishing the stability of individual differences via test-retest designs and comparison across a battery of multisensory tasks.

*4.2. Underlying Mechanisms of Multisensory Enhancement*

There are multiple ways in which a sound enhances visual perception. First, multisensory enhancement can result from mechanisms related to attention and response bias, where the sound enhances visual perception by reducing spatial and/or temporal uncertainty of the visual stimulus. For example, in previous studies where the onset of the visual target is unpredictable, the presence of a sound could serve as a temporal warning cue and enhance detection performance [42,43]. Such multisensory enhancement effects disappear when the visual target is presented without temporal ambiguity. Alternatively, the sound can reduce spatial uncertainty by orienting observers' attention towards a particular location. This is supported by work showing that involuntary orienting of spatial attention induced by a lateralized sound enhances early visual perceptual processing [44,45], leading to enhanced perceived visual contrast [46], and improved discrimination performance [47], even when the sound is not predictive of visual target location.

However, it is unlikely that the multisensory enhancement by IP sound relative to BL or OP conditions in this study is contributed by the reduction of spatial and/or temporal uncertainty. This is because the visual target was always presented at a fixed temporal interval from when fixation disappeared. In other words, there is little ambiguity regarding the temporal onset of the visual target. Similarly, the sound adopted in this study was not lateralized because it was presented via both speakers, thus not contributing to a spatial orienting effect. We speculate that this multisensory enhancement by IP sounds originates from perceptual enhancement. This account is supported by previous studies revealing early multisensory modulation of neural activity (probed by EEG/MEG (e.g., [37,48–50])), enhanced neural response in primary sensory cortices [5,10,48,51], and/or increased neural synchronization across neural areas in the multisensory network [49] (see review [52]). Interestingly, we did not find a suppression of the OP sound relative to BL condition. We expected a suppression effect based on previous literature (e.g., [53]). On the contrary, we found a significant enhancement of OP sound relative to BL after aligning data with individual observers' optimal auditory salience. Why is it the case? Despite not being temporally coincident with the visual stimulus, the OP sound might provide multisensory benefits relative to no sound by increasing the general arousal level of observers (similar to a transient boosting as proposed by [54]). The neural mechanism underlying IP and OP enhancement relative to no sound is to be unraveled by future neuroimaging work.

## 5. Conclusions

In sum, we parametrically manipulated the stimulus effectiveness of the task-irrelevant auxiliary sense and showed that it interacts with temporal phase coherence in modulating the strength of multisensory enhancement of visual contrast. Our approach is different from most previous studies where stimulus effectiveness of only one sense (usually the task-relevant sense) is manipulated (e.g., [19] used a range of effectiveness levels for auditory and one effectiveness level for visual) and other studies where stimulus effectiveness of the two senses is always matched (e.g., [20] used high/high vs.

low/low visual/auditory effectiveness). We found that there is an optimal auditory (auxiliary) stimulus effectiveness to induce the largest multisensory effect on visual (task-relevant) contrast at threshold, supporting that the *correspondence of stimulus effectiveness* across sensory inputs may be important to determine the magnitude of the multisensory effects. Taken together, discrepancies in the literature about the effect of stimulus effectiveness (e.g., [18,55]) could be driven by individual differences in the optimal stimulus effectiveness required for both the task-relevant and auxiliary sense as well as their level of correspondence with each other. Such individual differences could be a result of, as well as a contributing factor to, the 'context' supporting multisensory interactions [56] which awaits further investigation.

**Supplementary Materials:** The following are available online at http://www.mdpi.com/2411-5150/4/1/12/s1, Figure S1. Threshold, slope, and RT difference across auditory salience levels. Green: Enhancement group; Orange: Suppression group; Figure S2. Threshold, slope, and RT difference aligned to individual participants' optimal auditory salience; Table S1. Results of overall statistical analysis; Table S2. Results of statistical analysis based on data aligned by optimal auditory salience.

**Author Contributions:** Conceptualization, H.M.C. and V.M.C.; methodology, H.M.C. and V.M.C.; software, H.M.C. and V.M.C.; formal analysis, H.M.C., X.L., and V.M.C.; investigation, H.M.C. and X.L.; resources, V.M.C.; data curation, H.M.C., X.L., and V.M.C.; writing—original draft preparation, H.M.C. and X.L.; writing—review and editing, H.M.C., X.L. and V.M.C.; visualization, H.M.C. and X.L.; supervision, V.M.C.; project administration, V.M.C.; funding acquisition, V.M.C. All authors have read and agreed to the published version of the manuscript.

**Funding:** UMB Proposal Development Award (vc) and UMB undergraduate research funds (xl).

**Acknowledgments:** The authors thank a valuable team of undergraduate research apprentices who helped with data collection.

**Conflicts of Interest:** The authors declare no conflict of interest.

**Data Availability:** The datasets generated and/or analyzed in the current study are available in OSF repository: https://osf.io/cgwu4/?view_only=92465fc7f28049e6be763f7a8bdc7313.

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
