# Peer review of "Individual Differences in Multisensory Interactions: The Influence of Temporal Phase Coherence and Auditory Salience on Visual Contrast Sensitivity"

_2411-5150_

Round 1
Reviewer 1 Report
Overall a very interesting paper. I only have a few comments that I hope will help clarify the paper.
1) Apparatus and Stimuli lines 111 - 119 -> I would inverse the order of presentation here. As I read it the last sentence describes the settings based on individual performance. As such I think it reads easier for the reader if this is first (at least as an explanation) before the author(s) say how the ranges were selected.
2) Procedure lines 136 - 138 -> So I hate to sound like a dolt here but I am confused. A fixation point to me is a dot (hence a point) yet the "dimension range" was "0.913 x 0.913 - 2.931 x 2.931" (simplified math is: 0.913 x -2.018 x 2.931) which is a 3 dimensional coordinate (x, y, and z). Then the point changes it's shape, size, color, or angle every 150 ms? Not standard to me so a bit of explanation would be great.
3) Procedure lines 145 - 149 -> "..a brief cartoon video...." ? What does this mean? Also if incorrect there was no indicator other than a black box correct?
4) Figure 2 -> I would reduce the size of the points so that all could be seen, currently once can see 5 green, 3 red, and only 2 blue which makes the curve fitting hard to see.
Nice paper.
Author Response
Reply to comments from the reviews of “Individual differences in multisensory interactions: The influence of temporal phase coherence and auditory salience on visual contrast sensitivity”, which was submitted for publication in the Special Issue on the Multisensory Modulation of Vision for the journal Vision. We thank the editor and reviewers for their careful reading of our manuscript and thoughtful comments and suggestions, which have made the manuscript clearer. Below is a point-by-point reply for each change made based on reviewer concerns.
Reviewer #1
Overall a very interesting paper. I only have a few comments that I hope will help clarify the paper.
Apparatus and Stimuli lines 111 - 119 -> I would inverse the order of presentation here. As I read it the last sentence describes the settings based on individual performance. As such I think it reads easier for the reader if this is first (at least as an explanation) before the author(s) say how the ranges were selected.
We thank the reviewer for this suggestion and have reordered the presentation of information in the Apparatus and Stimuli section (p.6 in the manuscript word document). In the revised manuscript, we now first introduce the rationale of selecting contrast levels based on individual performance, then describe how we selected the contrast levels, and lastly provide the distribution of participants in each difficulty level.
Procedure lines 136 - 138 -> So I hate to sound like a dolt here but I am confused. A fixation point to me is a dot (hence a point) yet the "dimension range" was "0.913 x 0.913 - 2.931 x 2.931" (simplified math is: 0.913 x -2.018 x 2.931) which is a 3 dimensional coordinate (x, y, and z). Then the point changes it's shape, size, color, or angle every 150 ms? Not standard to me so a bit of explanation would be great.
We see the possible confusion, as stated by the reviewer, and have now clarified our description of the fixation stimulus used in our study (p.7 in the manuscript). To elaborate, we introduced changes in fixation stimulus (in size, shape, color, etc.) to capture participants’ attention to the center of the screen. For example, the fixation point varied in size between 0.913° and 2.931° in width/height. This is not a standard procedure in adult psychophysics but we designed this paradigm to be used throughout development, and this changing fixation stimulus is optimized for drawing attention to screen center in young infants. Also, it is important to note that the fixation stimulus disappeared when the visual target was presented; thus, the changing fixation stimulus should not interfere with task performance.
Procedure lines 145 - 149 -> "..a brief cartoon video...." ? What does this mean? Also if incorrect there was no indicator other than a black box correct?
Using a cartoon video as feedback/reward is another aspect of our experimental design optimized for using this paradigm to studying multisensory processing throughout development. The purpose of this is to reward orienting behavior, particularly in young infants, towards the visual stimulus. We clarified the presentation of the reward in the manuscript (p.7 of the manuscript). To elaborate here, the black box was presented at the correct location (where the visual target was presented) for both correct and incorrect responses. However, the cartoon video was only presented within the black box when the participant made a correct response.
Figure 2 -> I would reduce the size of the points so that all could be seen, currently once can see 5 green, 3 red, and only 2 blue which makes the curve fitting hard to see.
Thank you for the suggestion. We have changed the size of overlapping data points in Figure 2 to make them easier to distinguish and added text to the figure legend to avoid confusion regarding the meaning of differences in the size of data points: “Size of individual dots is adjusted so that overlapping data points are more visible.”
Nice paper.
Reviewer 2 Report
This paper examines the interactions between auditory intensity and temporal coherence manipulations on visual contrast thresholds. The main finding is lack of a cross-modal interaction that is consistent across participants. The core positive findings are based on exploratory analyses and as such, likely have a decent probability of being false positives. It is not clear that either of the individual differences are stable or meaningful. Ideally, there would be a follow-up study in which the subject classification rules and max interaction rules are pre-determined. Further it would be more informative if a subject’s interaction direction and max interaction level were stable across testing sessions and/or related to any measurements external to the task.
Minor points
Pg 3, Ln 120: Specify that sounds were the same on each speaker in the methods. It is clear from elsewhere that the auditory cues were not spatially relevant, but it should be specified here.
Section 2.3: Make it more explicit that the task is two-alternative forced choice and that only eye-movement response were collected.
Pg.6, Ln 195-196: Were the averages for outlier detection within condition? If not, it would bias the slowest condition results to appear faster.
Pg 6, ln 206: Bimodality is notoriously difficult to test for. What made it obvious? Was zero the midpoint of the modes?
Section 3.1: Were the subject exclusion criteria determined a priori
Author Response
Reply to comments from the reviews of “Individual differences in multisensory interactions: The influence of temporal phase coherence and auditory salience on visual contrast sensitivity”, which was submitted for publication in the Special Issue on the Multisensory Modulation of Vision for the journal Vision. We thank the editor and reviewers for their careful reading of our manuscript and thoughtful comments and suggestions, which have made the manuscript clearer. Below is a point-by-point reply for each change made based on reviewer concerns.
Reviewer #2
This paper examines the interactions between auditory intensity and temporal coherence manipulations on visual contrast thresholds. The main finding is lack of a cross-modal interaction that is consistent across participants. The core positive findings are based on exploratory analyses and as such, likely have a decent probability of being false positives. It is not clear that either of the individual differences are stable or meaningful. Ideally, there would be a follow-up study in which the subject classification rules and max interaction rules are predetermined. Further it would be more informative if a subject’s interaction direction and max interaction level were stable across testing sessions and/or related to any measurements external to the task.
We thank the reviewer for raising this important concern. We agree that our exploratory analyses should be replicated to determine stability. For this reason, we have now included the following explanation in the Discussion:
“ It is important to note that these results are based on exploratory analysis, to be confirmed by future studies where these classification methods of participants and/or stimulus intensity based on performance are pre-determined.” (p. 18 in the manuscript).
“We encourage future work to examine factors contributing towards individual differences of multisensory enhancement effects, as well as to establish the stability of these individual differences via test-retest designs and comparison across a battery of tests related to multisensory processing .” (p. 19 in the manuscript).
However, we would like to have the opportunity to publish the manuscript without additional confirmatory data for three reasons: (1) We attempted two kinds of exploratory analyses (subject classification rules and max interaction rules, as succinctly labeled by Reviewer #2) which yielded similar conclusion that the multisensory effect changes based on auditory intensity levels. (2) We have explicitly described that these analyses were exploratory throughout the manuscript so that the readers are informed. (3) One key purpose of this paper is to increase awareness about individual differences in multisensory effects. Sharing these exploratory analyses to account for individual differences to the field will allow these rules to be pre-determined and verified in future studies by not only us, but also by others.
Minor points
Pg 3, Ln 120: Specify that sounds were the same on each speaker in the methods. It is clear from elsewhere that the auditory cues were not spatially relevant, but it should be specified here.
We have incorporated the reviewer’s suggestion. Thank you. (p.8 in the manuscript)
Section 2.3: Make it more explicit that the task is two-alternative forced choice and that only eye-movement response were collected.
We have added a sentence at the beginning of the Procedure (section 2.3) to introduce the task and response method: “This experiment adopted a two-alternative forced-choice paradigm to probe visual contrast threshold based on participants’ eye movement response (left or right).”
Pg.6, Ln 195-196: Were the averages for outlier detection within condition? If not, it would bias the slowest condition results to appear faster.
We thank the reviewer for raising this valid concern. Outliers were detected within condition (in-phase, out-phase, baseline) and within auditory intensity. We have clarified this in the text (p.10 in the manuscript).
Pg 6, ln 206: Bimodality is notoriously difficult to test for. What made it obvious? Was zero the midpoint of the modes?
The bimodal nature of our distribution can be seen in the graphical density distribution of threshold differences (BL-IP) at the medium auditory salience level (39.5 dB) (see attachment). The midpoint of the modes was close to zero. We can include this figure in the manuscript if the reviewer thinks it would be helpful.
Section 3.1: Were the subject exclusion criteria determined a priori
No, subject exclusion criteria were not determined a priori, i.e., before data collection, but were informed by graphical distribution of the data before statistical analysis. We have clarified this in the manuscript (p. 11 in the manuscript).
